# Projected Off-Policy Q-Learning (POP-QL) for Stabilizing Offline Reinforcement Learning

## Abstract

A key problem in off-policy Reinforcement Learning (RL) is the mismatch, or *distribution shift*, between the dataset and the distribution over states and actions visited by the learned policy. This problem is exacerbated in the fully offline setting. The main approach to correct this shift has been through importance sampling, which leads to high-variance gradients. Other approaches, such as conservatism or behavior-regularization, regularize the policy at the cost of performance. In this paper, we propose a new approach for stable off-policy Q-Learning that builds on a theoretical result by Kolter (2011). Our method, Projected Off-Policy Q-Learning (POP-QL), is a novel actor-critic algorithm that simultaneously reweights off-policy samples and constrains the policy to prevent divergence and reduce value-approximation error. In our experiments, POP-QL not only shows competitive performance on standard benchmarks, but also out-performs competing methods in tasks where the data-collection policy is significantly sub-optimal.

## 1 Introduction

Temporal difference (TD) learning is one of the most common techniques for reinforcement learning (RL). Compared to policy gradient methods, TD methods tend to be significantly more data-efficient. One of the primary reasons for this data-efficiency is the ability to perform updates off-policy, where policy updates are performed using a dataset that was collected using a different policy. Unfortunately, due to the differences in distribution between the off-policy and on-policy datasets, TD methods that employ function approximation may diverge or result in arbitrarily poor value approximation when applied in the off-policy setting (Sutton & Barto, 2020, p. 260). This issue is exacerbated in the fully offline RL setting, where the training dataset is fixed and the agent must act off-policy in order to achieve high performance. Figure 1 illustrates this *distribution shift* issue on the Frozen Lake toy problem. In this example, we perform Q-Learning with a linear function approximate on a fixed dataset collected with one policy, the "data-collection" policy, to evaluate another policy, the "evaluation" policy. We can see that vanilla Q-learning diverges as the dataset shifts further off-policy.

Nearly all methods for addressing off-policy distribution shift fall into two categories: importance sampling methods that reweight samples from the dataset to approximate the on-policy distribution, or regularization methods that minimize the distribution shift directly or through penalizing the value function in low-support areas. The former may lead to extremely high variance gradient updates, and the latter only works well when the data-collection policy is close to optimal, which is not the case in most real-world datasets.

An alternative approach has been suggested by Kolter (2011), who provide a contraction mapping condition that, when satisfied, guarantees convergence of TD-learning to a unique fixed point. They propose to project the sampling distribution onto this convex condition and thereby significantly reduce approximation error. Figure 1 shows that when this contraction mapping condition is (approximately) satisfied, Q-learning converges with low approximation error. However, this approach does not scale to modern RL tasks, because it invokes a batch semi-definite programming (SDP) solver to solve the projection for each batch update.

**Contribution**    In this paper, we build on Kolter (2011)'s theoretical contribution and propose Projected Off-Policy Q-Learning (POP-QL), which makes two significant improvements over previous work. First, we consider a new sampling projection that allows for a more computationally efficient algorithm, since a closed-form solution exists for the inner optimization of the dual problem. This

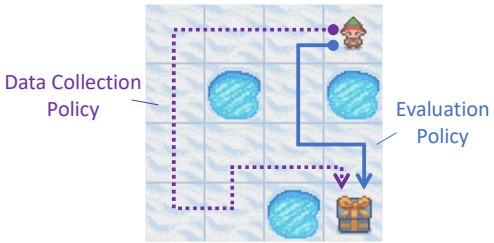 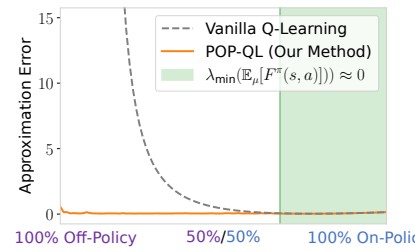

Figure 1: Off-policy evaluation on a simple grid environment, "Frozen Lake". The goal of this task is to evaluate a policy (—) from a suboptimal data policy (···) that is $\epsilon$-dithered for sufficient coverage ($\epsilon = 0.2$). The right plot shows approximation error from a linear Q-function trained using Vanilla Q-Learning and POP-QL (our method) with a dataset interpolated between off-policy and on-policy. Unlike Vanilla Q-Learning, POP-QL avoids divergence by projecting the sampling distribution $\mu$ onto the convex constraint $\mathbb{E}_{(s,a)\sim\mu}[F^\pi(s,a)] \succeq 0$, which enforced that the contraction mapping condition holds. The shaded region (■), indicates when the dataset approximately satisfies this condition (without reweighting); specifically, where $\lambda_{\min}(\mathbb{E}_{(s,a)\sim\mu}[F^\pi(s,a)]) > -0.005$.

computational improvement allows us to extend the technique to high-dimensional deep RL problems. Secondly, we extend this projection to the MDP setting and propose a new Q-Learning algorithm that jointly projects the policy and sampling distribution. Our proposed algorithm can plug into any Q-learning algorithm (such as Soft Actor Critic (Haarnoja et al., 2018)), significantly reducing the approximation error of the Q-function and improving policy performance. We evaluate our method on a variety of offline RL tasks and compare its performance to other offline RL approaches. Although, in its current iteration, our method struggles to reach state-of-the-art performance on tasks with near-expert data-collection policies, POP-QL is able to outperform many other methods when the data-collection policies are far from optimal (such as the "random" D4RL tasks). Most importantly, our results illustrate the power of the contraction mapping condition and the potential of this new class of offline RL techniques.

## 2 Background and Related Work

Off-policy TD learning introduces two distinct challenges that can result in divergence, which we refer to as *support mismatch* and *projected-TD instability*. Support mismatch, where the support of the data distribution differs from the on-policy distribution, can result in large Q-function errors in low-support regions of the state-action space. These Q-function errors can lead to policies that seek out regions of the state-action space for which the Q-function is overly-optimistic, thus leading to an increased support mismatch. This positive feedback loop can quickly diverge. In the online setting, this problem can be solved by frequently resampling data. However, in the fully offline setting, specific techniques are required to avoid such divergences, which we describe below.

The other challenge is instability inherent to off-policy TD learning. First described by Tsitsiklis & Van Roy (1996), the use of TD learning, function approximation, and off-policy sampling together, known as the *deadly triad* (Sutton & Barto, 2020, p. 264), can cause severe instability or divergence. This instability is caused by projecting TD-updates onto our linear basis[1], which can result in TD-updates that increase value error and, in some cases, diverge. See Appendix A for a three-state example of this divergence.

There are many methods that address the challenges of off-policy RL, most of which fall into two main categories. The first is importance sampling (IS). First proposed by Precup (2000), IS methods for RL approximate the on-policy distribution by reweighting samples with the ratio of the on-policy and data distribution densities. The challenge with this approach is the high variance in the updates of the re-weighting terms, which grows exponentially with the trajectory length. Many approaches have looked at methods for reducing this variance (Hallak & Mannor, 2017; Gelada & Bellemare, 2019; Nachum et al., 2019a;b; Liu et al., 2018; Lee et al., 2021). Emphatic-TD (Sutton et al., 2016; Zhang et al., 2020; Jiang et al., 2021) and Gradient-TD (Sutton et al., 2008) are other importance

---

[1]This challenge remains present in the deep RL setting.

sampling approaches that are provably stable in the full-support linear case. One critical challenge with IS methods is that they do not address support mismatch and, thus, tend to perform poorly on larger scale problems.

The second category of methods involves regularizing the policy towards the data-policy. This policy regularization can be done explicitly by ensuring the learned policy remains "close" to the data collection policy or implicitly through conservative methods. Explicit policy regularization can be achieved by penalizing KL-divergence between the learned policy and data policy (Fujimoto et al., 2019; Wu et al., 2019) or by regularizing the policy parameters (Mahadevan et al., 2014). However, even in small-scale settings, policy-regularization methods can be shown to diverge (Manek & Kolter, 2022). Conservative methods, on the other hand, involve making a conservative estimate of the value function, thus creating policies that avoid low-support regions of the state-action space. In the online learning setting, one of the most common conservative methods is Trust Region Policy Optimization (TRPO) (Schulman et al., 2015). However, TRPO does not extend well to the fully offline setting since the value estimates tend to be overly conservative as the learned policy diverges from the data policy. One of the most successful algorithms for offline RL is Conservative Q-Learning (CQL, Kumar et al. (2020b)), which adds a cost to out-of-distribution actions. Other methods use conservative value estimates with ensembles (Kumar et al., 2019) or model-based approaches (Yu et al., 2020; Kidambi et al., 2020). One downside to policy regularized methods (explicit or implicit) is that regularization can reduce policy performance when the data policy is sub-optimal, which we demonstrate in our experiments.

There are a few notable approaches that do not fit into either of these categories. Kumar et al. (2020a) present DisCor, which reweights the sampling distribution such that the TD fixed point solution minimizes Q-approximation error. However, approximations are needed to make this approach tractable. Additionally, this algorithm does address the support mismatch problem.

Another approach is TD Distribution Optimization (TD-DO) (Kolter, 2011), which seeks to reweight the sampling distribution such that the TD updates satisfy the contraction mapping condition, thereby ensuring that TD updates converge. Unfortunately, the use of this approach has been limited because it does not scale to modern RL tasks. This is due to the need to run a batch SDP solver to solve the associated distributional optimization task for each batch update. Our method, Projected Off-Policy Q-Learning (POP-QL), builds off this approach. In this work, we propose a new projection that easily scales to larger domains and extend our method to policy optimization, creating a novel algorithm that addresses both the *support mismatch* and *projected-TD instability* of off-policy RL.

# 3 Preliminaries and Problem Setting

In this work, we consider learning a policy that maximizes the cumulative discounted reward on a Markov Decision Process (MDP) defined as the tuple $(\mathcal{S}, \mathcal{A}, p, r, \gamma)$, where $\mathcal{S}$ and $\mathcal{A}$ are the state and action spaces, $p(\cdot|s, a)$ and $r(s, a)$ represent the transition dynamics and reward functions, and $\gamma \in [0, 1)$ is the discount factor. We approximate the action-value function as $Q(s, a) \approx w^\top \phi(s, a)$, where $\phi : \mathcal{S} \times \mathcal{A} \to \{x \in \mathbb{R}^k : \|x\|_2 = 1\}$ is a normalized basis function and $w \in \mathbb{R}^k$ are the parameters of the final, linear layer.

In the off-policy setting, we assume the agent cannot directly interact with the environment and instead only has access to samples of the form $(s, a, r(s, a), s')$, where $s' \sim p(\cdot|s, a)$ and $(s, a) \sim \mu$ for some arbitrary sampling distribution $\mu$. Because we can assume a fixed policy for much of our derivations and theory, we can simplify the math significantly by focusing on the finite Markov Reward Process setting instead.

## 3.1 Simplified Setting – Finite Markov Reward Process (MRP)

Consider the finite $n$-state Markov Reward Process (MRP) $(\mathcal{S}, p, r, \gamma)$, where $\mathcal{S}$ is the state space, $p : \mathcal{S} \times \mathcal{S} \to \mathbb{R}_+$ and $r : \mathcal{S} \to \mathbb{R}$ are the transition and reward functions, and $\gamma \in (0, 1)$ is the discount factor. [2] Because the state-space is finite, it can be indexed as $\mathcal{S} = \{1, \dots, n\}$, which allows us to use matrix rather than operator notation. In matrix notation, we use matrices $P$ and $R$, to represent the functions $p$ and $r$, where each row corresponds to a state. The value function associated with the MRP

---

[2] Note that, given a fixed policy, an MDP reduces to an MRP.

is the expected $\gamma$-discounted future reward of being in each state $V(s) := \mathbb{E}\left[\sum_{t=0}^{\infty}\gamma^{\top}r(s_t)\middle| s_0 = s\right]$. The value function is consistent with Bellman's equation in matrix form,

$$V = R + \gamma PV. \tag{1}$$

We approximate the value function as $V(s) \approx w^{\top}\phi(s)$, where $\phi : \mathcal{S} \to \{x \in \mathbb{R}^k : \|x\|_2 = 1\}$ is a fixed normalized basis function and we estimate parameters $w \in \mathbb{R}^k$. In matrix notation, we write this as $V \approx \Phi w$. In the off-policy setting, the sampling distribution $\mu$ differs from the stationary distribution $\nu$. In this setting, the temporal difference (TD) solution is the fixed point of the projected Bellman equation:

$$\Phi w^{\star} = \Pi_{\mu}(R + \gamma P\Phi w^{\star}), \tag{2}$$

where $\Pi_{\mu} = \Phi(\Phi^{\top}D_{\mu}\Phi)^{-1}\Phi^{\top}D_{\mu}$ is the projection onto the column space of $\Phi$ weighted by the data distribution $\mu$ through the matrix $D_{\mu} = \mathrm{diag}(\mu)$. This projection may be arbitrarily far from the true solution so that the error may be correspondingly large. In practice, $w^{\star}$ is often computed using TD-learning, a process that starts from some point $w_0 \in \mathbb{R}^k$ and iteratively applies Bellman updates,

$$w_{t+1} = w_t - \lambda\mathbb{E}_{\mu}\left[\left(\phi(s)^{\top}w_t - r - \phi\left(s'\right)^{\top}w_t\right)\phi(s)\right]. \tag{3}$$

Unfortunately, in the off-policy setting, TD-learning is not guaranteed to converge.

## 3.2 Contraction Mapping Condition

A $\gamma$-contraction mapping[3] is any function, $f : \mathbb{R}^n \to \mathbb{R}^n$, such that for some distribution $\mu$ and any $x_1, x_2 \in \mathbb{R}^n$:

$$\|f(x_1) - f(x_2)\|_{\mu} \leq \gamma\|x_1 - x_2\|_{\mu}, \tag{4}$$

where $\gamma \in [0, 1)$ and $\|\cdot\|_{\mu}$ is the weighted 2-norm. A key property of contraction mappings is that iteratively applying this function to any starting point $x_0 \in \mathbb{R}^n$ converges to a unique fixed point $x^* = f(x^*)$. This principle is used to prove convergence of on-policy TD-learning.

Under on-policy sampling, $\mu = \nu$, the projected Bellman operator, $\Pi_{\mu}\mathcal{B}(x) = \Pi_{\mu}(R + \gamma Px)$, is a contraction mapping.

$$\|\Pi_{\mu}\mathcal{B}(\Phi w_1) - \Pi_{\mu}\mathcal{B}(\Phi w_2)\|_{\mu} \leq \gamma\|\Phi w_1 - \Phi w_2\|_{\mu} \quad \forall w_1, w_2 \in \mathbb{R}^k. \tag{5}$$

Tsitsiklis & Van Roy (1996) use this property to both prove that on-policy TD Q-learning learning converges to a unique point and bound the approximation error of the resulting fixed point (Tsitsiklis & Van Roy, 1996, Lemma 6). However, in the off-policy setting with $\mu \neq \nu$ this property does not always hold. In fact, this condition can be violated even in MRPs with very small state spaces, see Appendix A for an example. Thus, the TD updates are not guaranteed to converge and can diverge under some off-policy sampling distributions.

To get around this challenge, Kolter (2011) proposed a new approach. First, they transformed the contraction mapping condition into a linear matrix inequality (LMI) through algebraic manipulation:

$$\mathbb{E}_{s\sim\mu}[F(s)] \succeq 0, \text{ where } F(s) = \mathbb{E}_{s'\sim p(\cdot|s)}\left[\begin{bmatrix}\phi(s)\phi(s)^{\top} & \phi(s)\phi(s')^{\top} \\ \phi(s')\phi(s)^{\top} & \phi(s)\phi(s)^{\top}\end{bmatrix}\right]. \tag{6}$$

We provide a derivation of this LMI in Appendix B.1. Using this formulation, they present an algorithm to find a new sampling distribution that satisfies this contraction mapping condition and proves a bound on the approximation error of their approach. Unfortunately, this method scales poorly because it requires solving an SDP problem alongside each batch update. Thus, the method remains impractical for the deep RL tasks, and has seen virtually no practical usage in the years since.

# 4 Projected Off-Policy Q-Learning (POP-QL)

Our method, Projected Off-Policy Q-Learning (POP-QL), is also centered on the contraction mapping condition (Eq. (6)). However, unlike previous work, we propose a new method that significantly

---

[3]A contraction mapping can be defined for any metric space, but here we focus on the metric space defined by the Euclidean space and weighted Euclidean metric.

improves the computational cost of the projection, allowing POP-QL to scale to large-scale domains. Additionally, we introduce a new policy optimization algorithm that simultaneously projects the policy and sampling distribution in order to satisfy the contraction mapping condition. This policy optimization algorithm allows POP-QL to address both the *support mismatch* and *projected-TD instability* challenges of off-policy Q-learning.

We start by deriving the POP-QL reweighting procedure in the finite MRP setting under a fixed policy and later extend our method to the MDP setting together with policy regularization.

## 4.1 POP-QL on Markov Reward Processes

In the MRP setting (or the fixed-policy setting), the goal of POP-QL is to compute a new sampling distribution that satisfies the contraction mapping condition in Eq. (6) and thus stabilizes off-policy training. However, if the target distribution differs significantly from the source distribution $\mu$, this can result in large reweighting factors, which can decrease the stability of the training process. Thus, we are looking for the "closest" distribution that satisfies Eq. (6). Unlike Kolter (2011), we propose to use the I-projection instead of the M-projection, which allows us to find an analytical solution to the inner part of the Lagrangian dual and thereby significantly simplifies the problem. Additionally, we argue in Appendix C.1 that this choice is more suitable for the RL setting. We first formulate the problem as minimizing the KL divergence between the data distribution $\mu$ and a reweighted distribution $q$ such that TD update is stable under $q$,

$$\underset{q}{\text{minimize}}\, D_{\mathrm{KL}}(q \,\|\, \mu) \quad \text{s.t.} \quad \mathbb{E}_{s \sim q}[F(s)] \succeq 0. \tag{7}$$

The corresponding unconstrained dual problem based on a Lagrange variable $Z \in \mathbb{R}^{2k}$ is given by

$$\underset{Z \succeq 0}{\text{maximize}}\, \underset{q}{\text{minimize}}\, D_{\mathrm{KL}}(q\|\mu) - \operatorname{tr} Z^{\top} \mathbb{E}_q[F(s)]. \tag{8}$$

The solution to this dual problem is equal to the primal problem under strong duality, which holds in practice due to the fact that this corresponds to a convex optimization problem. Now, consider the inner optimization problem over $q$ in Eq. (8). This optimization problem can be rewritten as $\text{minimize}_q\ -H(q) - \mathbb{E}_q\left[\log \mu(s) + \operatorname{tr} Z^{\top} F(s)\right]$, which has a simple analytical solution:

$$q^{\star}(s) \propto \exp\left(\log \mu(s) + \operatorname{tr} Z^{\top} F(s)\right) = \mu(s) \exp\left(\operatorname{tr} Z^{\top} F(s)\right). \tag{9}$$

Notice that our target distribution $q^{\star}$ is simply a reweighting of the source distribution $\mu$ with weights $\exp\left(\operatorname{tr} Z^{\top} F(s)\right)$. To compute the weights, we need to solve for the Lagrange variable $Z$. Plugging the analytical solution for $q^{\star}$ back into Eq. (8) yields

$$\underset{Z \succeq 0}{\text{minimize}}\, \mathbb{E}_{\mu}\left[\exp\left(\operatorname{tr} Z^{\top} F(s)\right)\right]. \tag{10}$$

In practice, we minimize over the set $Z \succeq 0$ by re-parametrizing $Z$ as

$$Z = \begin{bmatrix} A \\ B \end{bmatrix} \begin{bmatrix} A \\ B \end{bmatrix}^{\top} \tag{11}$$

where $A, B \in \mathbb{R}^{k \times k}$. This formulation ensures that $Z$ is positive semi-definite, $Z \succeq 0$, for any $A$ and $B$. Thus, we can directly optimize over $A$ and $B$ and ignore the positive semi-definite condition. With this formulation for $Z$ and plugging the definition of $F$ (Eq. (6)) we can rewrite the dual optimization problem (Eq. (10)) as:

$$\underset{A,B}{\text{minimize}}\, \mathbb{E}_{\mu}\left[\exp\left(\|A^{\top}\phi(s)\|_2^2 + \|B^{\top}\phi(s)\|_2^2 + 2\mathbb{E}_{s' \sim p(\cdot|s)}\left[\langle B^{\top}\phi(s), A^{\top}\phi(s')\rangle\right]\right)\right] \tag{12}$$

Solving for matrices $A, B$ yields the I-projected sampling distribution $q^*$ according to Eq. (9).

## 4.2 Extension to Markov Decision Processes

The theory presented in the previous section can be extended to the MDP setting through a simple reduction to an MRP. In a MDP, we also have to consider the action space, $\mathcal{A}$, and the policy, $\pi$. In this setting, our contraction mapping LMI becomes

$$\mathbb{E}_{(s,a) \sim q}[F^{\pi}(s,a)] \succeq 0, \tag{13}$$

---

**Algorithm 1** Projected Off-Policy Q-Learning (POP-QL)

---

Initialize: feature function $\phi_{\theta^\phi}$, Q-function parameters $w$, policy $\pi_{\theta^\pi}$,
       g-function $g_{\theta^g}$, and Lagrange matrices $A$ and $B$.

**for** step $t$ in $1, \ldots, N$ **do**

   $(s, a, r, s')_{1,\ldots,m} \sim \mu$                     ▷ Sample minibatch from dataset

   $\tilde{a} \sim \pi_{\theta^\pi}(s) \; \tilde{a}' \sim \pi_{\theta^\pi}(s')$           ▷ Sample new actions from policy

   $q_{\text{target}} := r + \gamma w^\top \phi_{\theta^\phi}(s', \tilde{a}')$         ▷ Compute Q-function target value

   $y_A, y_B, y'_A := A^\top \phi_{\theta^\phi}(s, a), B^\top \phi_{\theta^\phi}(s, a), A^\top \phi_{\theta^\phi}(s', a')$     ▷ Compute dual values

   $u := \exp\left(\|y_A\|_2^2 + \|y_B\|_2^2 + 2g_{\theta^g}(s, a)\right)/\bar{u}$     ▷ Compute minibatch-normalized weight

   $A, B \leftarrow [A, B] - \lambda_{A,B} u \nabla_{A,B} \left(\|y_A\|_2^2 + \|y_B\|_2^2 + 2\langle y_B, y'_A \rangle\right)$    ▷ Update Lagrange matrices

   $\theta^g \leftarrow \theta^g - \lambda_g \nabla_{\theta^g} \left(g_{\theta^g}(s, a) - \langle y_B, y'_A \rangle\right)^2$        ▷ Update g-function parameters

   $[\theta^Q, w] \leftarrow \theta^Q - \lambda_Q u \nabla_{\theta^Q, w} \left(w^\top \phi_{\theta^\phi}(s, a) - q_{\text{target}}\right)^2$    ▷ Update Q-function parameters

   $\theta^\pi \leftarrow \theta^\pi - \lambda_\pi \nabla_{\theta^\pi} \left(\mathcal{L}_Q + \alpha \mathcal{L}_{\text{entropy}} - \beta u \langle y_B, y'_A \rangle\right)$       ▷ Augment SAC policy loss

---

$$\text{where } F^\pi(s, a) = \mathbb{E}_{s' \sim p(\cdot | s, a), a' \sim \pi(s')} \left[\begin{bmatrix} \phi(s,a)\phi(s,a)^\top & \phi(s,a)\phi(s',a')^\top \\ \phi(s',a')\phi(s,a)^\top & \phi(s,a)\phi(s,a)^\top \end{bmatrix}\right]. \tag{14}$$

Using the idea that, given a fixed policy, any MDP reduces to an MRP, we extend Theorem 2 from Kolter (2011) to show that the TD-updates converge to a unique fixed point with bounded approximation error for any finite MDP where $\pi$ and $\mu$ satisfy this condition.

**Lemma 1.** *Let $w^\star$ be the least-squares solution to the Bellman equation for a fixed policy $\pi$:*

$$w^* = \arg\min_w \mathbb{E}_{(s,a)\sim\mu} \left[(\phi(s,a)^\top w - r(s,a) - \gamma \mathbb{E}_{s'\sim p(\cdot|s,a), a'\sim\pi(s')}\phi(s',a')^\top w)^2\right] \tag{15}$$

*and let $\mu$ be some distribution satisfying the MDP contraction mapping condition (Eq. (13)). Then*

$$\mathbb{E}_\mu \left[(\phi(s,a)^\top w^\star - V(s,a))^2\right] \leq \frac{1 + \gamma\sqrt{\delta(\nu,\mu)}}{1-\gamma} \min_w \mathbb{E}_\mu \left[(\phi(s,a)^\top w - V(s,a))^2\right], \tag{16}$$

*where $\nu$ is the stationary distribution, $\delta(\nu, \mu) = \max_{s,a,\tilde{s},\tilde{a}} \frac{\nu(s,a)}{\mu(s,a)} \cdot \frac{\mu(\tilde{s},\tilde{a})}{\nu(\tilde{s},\tilde{a})}$*

See Appendix B.2 for proof. As before, we are looking to project our sampling distribution to satisfy this condition. With this reduction, we rewrite the analytical solution for the projected sampling distribution from Eq. (9) as

$$q^\star(s, a) \propto \mu(s, a) \exp\left(\|y_A\|_2^2 + \|y_B\|_2^2 + 2\mathbb{E}_{s'\sim p(\cdot|s), a'\sim\pi(\cdot|s')}[\langle y_B, y'_A \rangle]\right) \tag{17}$$

where $y_A = A^\top \phi(s,a)$, $y_B = B^\top \phi(s,a)$, and $y'_A = A^\top \phi(s', a')$ and $A$ and $B$ are the solutions to the following optimization problem,

$$\underset{A,B}{\text{minimize}} \; \mathbb{E}_\mu \left[\exp\left(\|y_A\|_2^2 + \|y_B\|_2^2 + 2\mathbb{E}_{s'\sim p(\cdot|s)}[\langle y_B, y'_A \rangle]\right)\right] \tag{18}$$

Now, by Lemma 1 and assuming strong duality holds, the fixed point of the projected Bellman equation under the sampling $q^\star$ has bounded approximation error.

## 4.3 Practical Implementation

**Two-Time-Scale Optimization** Because there is an expectation inside an exponential, we must perform a two time-scale optimization to be able to use sample-based gradient descent. We introduce a new function approximator $g_\theta$ to approximate the inner expectation:

$$g_\theta(s, a) \approx \mathbb{E}_{s'\sim p(s'|s), a'\sim\pi(\cdot|s')}[\langle y_B, y'_A \rangle], \tag{19}$$

which can be optimized using gradient descent.

If we assume $g_\theta(s)$ has sufficient expressive power and has converged, we can estimate the gradient of our objective with respect to $A$ and $B$, $\nabla_{A,B}\mathbb{E}_\mu \left[\exp\left(\text{tr } Z^\top F(s)\right)\right]$, using samples from our sampling distribution, $\mu$:

$$\mathbb{E}_{s,a\sim\mu, s'\sim p(s'|s), a'\sim\pi(\cdot|s')} \left[u(s,a) \cdot \nabla_{A,B}\left(\|y_A\|_2^2 + \|y_B\|_2^2 + 2\langle y_B, y_A \rangle\right)\right] \tag{20}$$

Figure 2: Heat-maps of three state distributions for the "Frozen Lake" environment. On the left is the off-policy sampling distribution, on the right is the on-policy sampling distribution, and, in the middle, is the projection of off-policy sampling distribution onto the contraction mapping set (Eq. (6)) computed by POP-QL. Note that only a minor change to the off-policy sampling distribution is needed to satisfy the contraction mapping condition and, thus, guarantee convergence of TD-learning.

where $u(s, a)$ are the sample reweighting terms defined as:

$$u(s, a) = \exp\left(\|y_A\|_2^2 + \|y_B\|_2^2 + 2g_\theta(s)\right), \quad q^*(s, a) = u(s, a)\mu(s, a). \tag{21}$$

With this approximation, we can perform two-time-scale gradient descent. The gradient updates for $g_\theta$ and the Lagrange matrices $A$ and $B$ become

$$\theta \leftarrow \theta - \lambda_\theta \nabla_\theta \left(g_\theta(s, a) - \mathbb{E}_{s' \sim p(s'|s,a), a' \sim \pi(s')}\left[\langle y_B, y_A'\rangle\right]\right)^2,$$
$$A, B \leftarrow [A, B] - \lambda_{A,B}\mathbb{E}_{\mu,p}\left[u(s, a) \cdot \nabla_{A,B}\left(\|y_A\|_2^2 + \|y_B\|_2^2 + 2\langle y_B, y_A'\rangle\right)\right], \tag{22}$$

We also found it helpful to normalize the g-function by the spectral norm of $A$ and $B$. See Appendix C.2 for details.

**Low-Rank Approximation** Empirically, we found the solution to our Lagrange dual optimization problem is typically a low-rank matrix with rank $r \leq 4$; this is not surprising in hindsight: under the on-policy distribution the matrix $\mathbb{E}_{(s,a) \sim q}[F^\pi(s, a)]$ is *already* positive definite (a consequence of the fact that TD will converge on-policy), and so it is intuitive that this bound would only need to be enforced on a low-dimensional subspace, corresponding to a low-rank dual solution. Thus, we can substantially reduce the computational cost of the method by using Lagrange matrices $A, B \in \mathbb{R}^{k \times r}$, where $r = 4$. This is essentially akin to low-rank semi-definite programming (Burer & Monteiro, 2003), which has proven to be an extremely competitive and scalable approach for certain forms of semi-definite programs. Furthermore, while not a primary motivation for the method, we found this low-rank optimization improves convergence of the dual matrices. In total, this leads to a set of updates that are fully linear in the dimension of the final-layer features, and which ultimately presents a relatively modest increase in computational cost over standard Q-Learning.

## 4.4 Policy Optimization

So far, we have assumed a fixed policy in order to compute a sampling distribution and use that sampling distribution to compute a Q-function with low approximation error. Next, we need to find a policy that maximizes this Q-function. However, we want to avoid policies that result in very large reweighting terms for a couple reasons: 1) state action pairs with large reweighting terms correspond to low-support regions of the state-space, and 2) large reweighting terms increase the variance of the gradient updates of our Lagrange matrices. To keep these reweighting terms small, we jointly project $\pi$ and the sampling distribution $\mu$ using a balancing term $\beta \in \mathbb{R}_+$:

$$\underset{\pi, q}{\text{maximize}} \; \mathbb{E}_\mu[Q^\pi(s, a)] - \beta D_{\text{KL}}(q \,\|\, \mu) \quad \text{s.t.} \quad \mathbb{E}_{(s,a) \sim q}[F^\pi(s, a)] \succeq 0 \tag{23}$$

We can solve this optimization using the technique from the previous section. The gradient updates for $A$, $B$, and the g-function stay the same and the gradient updates for the policy become

$$\nabla_\pi \mathbb{E}_\mu[Q^\pi(s, a)] + 2\beta \mathbb{E}_{q^\pi}\left[\nabla_\pi \mathbb{E}_\pi\left[\langle y_B, y_A'\rangle\right]\right]. \tag{24}$$

See Appendix B.3 for a detailed derivation and Algorithm 1 for the pseudocode of our algorithm.

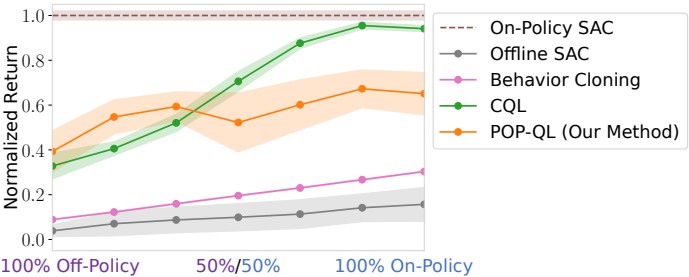

Figure 3: Offline policy optimization performance on the Frozen Lake domain (Fig. 1) averaged over 5 random seeds (shaded area is standard error). As before, we varied the datasets by interpolating between the dataset collected by the "data-collection policy" (···) and the dataset collected by the "evaluation policy" (—), both with $\epsilon$-dithering for sufficient coverage ($\epsilon = 0.2$). Both our POP-QL (our method) and CQL are able to find a policy that outperforms behavior cloning without diverging.

# 5 Experiments and Discussion

**Small Scale – Frozen Lake**    Frozen Lake is a small grid navigation task where the objective is to reach the goal state while avoiding the holes in the ice, which are terminal states. Figure 1 shows a visualization of the Frozen Lake environment. Note, unlike the standard Frozen Lake environment, our Frozen Lake has no terminal states. Instead, falling in the "holes" moves the agent back to the starting location. For tabular Q-learning, this is a very simple task. However, using function approximation with a linear function approximator can cause offline Q-Learning to quickly diverge.

We illustrate this divergence first with a policy evaluation task. In this task, the goal is to approximate the Q-function for an "evaluation" policy with data collected from a separate "data-collection" policy. We use a random featurize for the state-action space with dimension $k = 63$ (the true state-action space has a cardinality of $64$). We train a linear function approximator with both vanilla Q-learning and POP-QL as we linearly interpolate the dataset between 100% offline (meaning collected entirely from the "data collection" policy) and 100% (meaning collected entirely from the "evaluation" policy). Figure 1 shows a graph of the results. When trained offline, vanilla Q-learning quickly diverges, whereas POP-QL remains stable for all the datasets. We also note that the datasets for which vanilla Q-learning converges with low approximation error correspond to those that roughly satisfy our contraction mapping condition, exactly as our theory would predict. Figure 2 illustrates how the projection made by POP-QL changes the sampling distribution only slightly compared to exactly projecting onto the on-policy distribution (importance sampling).

We also perform a policy optimization task with the same datasets. In this task, the goal is to compute the policy with the highest return using offline data. We compare our method against online SAC, offline SAC, and CQL. Appendix D discusses the hyper-parameter tuning procedures for these baseline methods. Figure 3 shows the expected normalized returns of the policies computed using various methods.

**D4RL Tasks**    D4RL (Fu et al., 2020) is a standardized collection of offline RL tasks. Each task consists of an environment and dataset. The datasets for each task are collected by using rollouts of a single policy or a mixture of policies. The goal of each task is to learn a policy exclusively from these offline datasets that maximizes reward on each environment.

We compare our method against vanilla SAC (Haarnoja et al., 2018) run on offline data, Behavior Cloning, and CQL (Kumar et al., 2020b) (with the JaxCQL codebase (Geng, 2022). For each of these methods, we run for 2.5 million gradient steps. Using the JaxCQL codebase, we were not able to replicate the results of CQL with the hyper-parameters presented in Kumar et al. (2020b). Instead we performed our own small hyper-parameter sweep (details in Appendix D). For further comparison, we also include results for bootstrapping error reduction (BEAR) (Kumar et al., 2019), batch-constrained Q-learning (BCQ) (Fujimoto et al., 2019), and AlgaeDICE (Nachum et al., 2019b) reported in Fu et al. (2020). We looked at 2 categories of environments: 1) The OpenAI Gym (Brockman et al., 2016) environments Hopper, Half Cheetah, and Walker2D, and 3) the Franka Kitchen environments (Gupta et al., 2019). For each method, we used fixed hyper-parameters for environment category.

Table 1: Results on the D4RL MuJoCo offline RL tasks (Fu et al., 2020). We ran offline SAC (SAC-off), CQL, and POP-QL for 2.5M gradient steps. [†]Results reported from Fu et al. (2020). We can see our method, POP-QL, outperforms other methods on the very suboptimal datasets ("random"), but falls behind on the others.

|  | SAC-off | BEAR[†] | BCQ[†] | aDICE[†] | CQL | POP-QL |
|---|---|---|---|---|---|---|
| hopper-random | 11.88 | 9.50 | 10.60 | 0.90 | 0.92 | **20.43** |
| halfcheetah-random | 27.97 | 25.50 | 2.20 | -0.30 | -1.12 | **29.93** |
| walker2d-random | 4.55 | **6.70** | 4.90 | 0.50 | -0.02 | -0.31 |
| hopper-medium | 2.32 | 47.60 | **54.50** | 1.20 | 44.71 | 24.97 |
| halfcheetah-medium | 57.28 | 38.60 | 40.70 | -2.20 | **62.37** | 43.03 |
| walker2d-medium | 0.94 | 33.20 | **53.10** | 0.30 | 7.36 | 13.49 |
| hopper-medium-replay | 18.54 | **96.30** | 33.10 | 1.10 | 2.05 | 19.89 |
| halfcheetah-medium-replay | 43.34 | 38.60 | 38.20 | -0.80 | **50.03** | 34.99 |
| walker2d-medium-replay | 5.05 | 19.20 | 15.00 | 0.60 | **63.97** | 11.61 |
| hopper-medium-expert | 2.31 | 4.00 | **110.90** | 1.10 | 45.72 | 10.09 |
| halfcheetah-medium-expert | 6.41 | 51.70 | 64.70 | -0.80 | **82.76** | 51.21 |
| walker2d-medium-expert | 0.08 | 10.80 | **57.50** | 0.40 | 10.12 | 36.01 |

Table 2: Results on the D4RL kitchen offline RL tasks (Fu et al., 2020). We ran offline SAC (SAC-off), CQL, and POP-QL for 2.5M gradient steps. [†]Results reported by Fu et al. (2020). Our method, POP-QL, out-performs offline SAC and CQL, but falls behind BEAR and BCQ.

|  | SAC-off | BEAR[†] | BCQ[†] | aDICE[†] | CQL | POP-QL |
|---|---|---|---|---|---|---|
| kitchen-complete | 0.12 | 0.00 | **8.10** | 0.00 | 0.00 | 0.00 |
| kitchen-partial | 0.00 | 13.10 | **18.90** | 0.00 | 6.12 | 6.38 |
| kitchen-mixed | 0.00 | **47.20** | 8.10 | 0.00 | 0.62 | 1.56 |

Tables 1 and 2 show the results on these tasks. We can see that our method is competitive or outperforms all other methods on the "random" and "medium" datasets, but falls behind on the "medium-expert" environments. This is because our method, unlike most other offline methods, does not perform any regularization towards the data collection policy[4].

# 6    Conclusion and Future Directions

In this paper, we present Projected Off-Policy Q-Learning (POP-QL), a new method for reducing approximation errors in off-policy and offline Q-learning. POP-QL performs an approximate projection of both the policy and sampling distribution onto a convex set, which guarantees convergence of TD updates and bounds the approximation error.

Unlike most other offline RL methods, POP-QL does not rely on pushing the learned policy towards the data-collection policy. Instead POP-QL finds the smallest adjustment to the policy and sampling distribution that ensures convergence. This property is exemplified in our experiments, especially when the data-collection policies are significantly sub-optimal. In small-scale experiments, we show that our method significantly reduces approximation error of the Q-function. We also evaluate our method on standardized Deep RL benchmarks. POP-QL outperforms other methods when the datasets are far from the optimal policy distribution, specifically the "random" datasets, and is competitive but falls behind the other methods when the dataset distribution gets closer to the on-policy distribution.

This paper illustrates the power of the contraction mapping condition first introduced by Kolter (2011) for offline RL and introduces a new class of offline RL techniques. While, in its current iteration, this method does not outperform the state-of-the-art methods on every domain, our results suggest the exciting potential of this new technique. We think this reduced performance on some the D4RL tasks is primarily due to training instabilities introduced by the min-max optimization of the policy and Lagrange matrices. As with many other RL algorithms, finding implementation tricks, such as target Q-networks (Mnih et al., 2015) and double Q-networks (Hasselt et al., 2016; Fujimoto et al., 2018),

---

[4]The data collection policy is often called the *behavior policy*.

is critical to stabilizing learning. In future work, we hope to address the instability of the Lagrange matrix optimization, thus providing a method that consistently out-performs competing methods.

**Acknowledgements**

This research was sponsored by Robert Bosch GMBH under award number 0087016732PCRPO0087023984. The views and conclusions contained in this document are those of the author and should not be interpreted as representing the official policies, either expressed or implied, of any sponsoring institution, the U.S. government or any other entity.

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

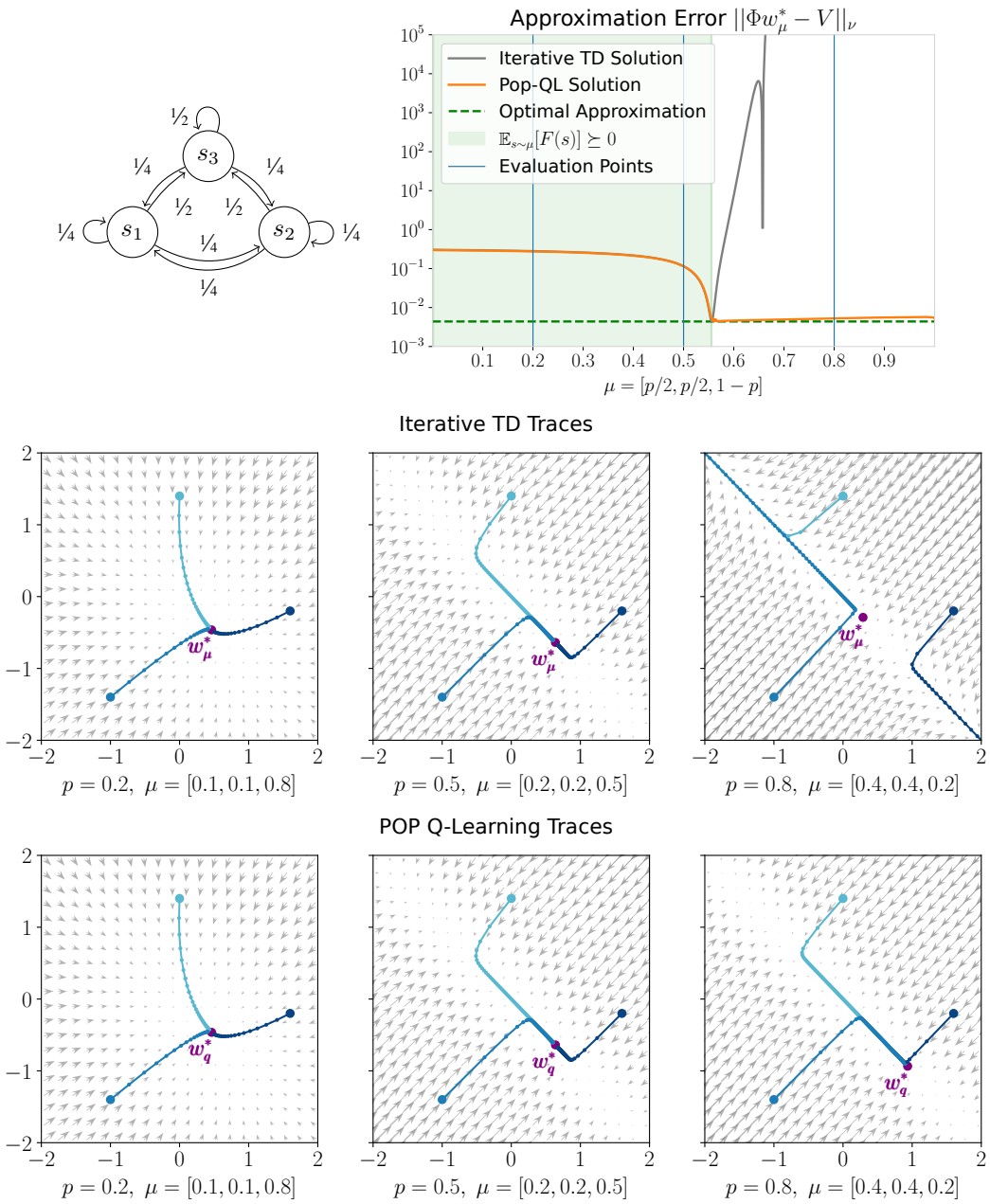

Figure 4: The three-state Markov process by Manek & Kolter (2022) (top-left), a plot of Q-function approximation error over different sampling distributions using Iterative TD and POP-QL (top-right), and TD traces for at three different evaluation sampling distributions. We can see that when the contraction mapping condition is satisfied ($p \lesssim 0.55$), the Iterative TD and POP-QL solutions are identical. However, when this condition is violated ($p > 0.55$), Iterative TD diverges, whereas POP-QL converges and retains a low approximation error.

# A    Off-Policy Contraction Mapping Example

To illustrate how the contraction mapping condition impacts TD updates, we use the simple three-state MRP introduced in Manek & Kolter (2022) (Fig. 4). In this example, the value function is given by $V = [1, \ 1, \ 1.05]^\top$, with discount factor $\gamma = 0.99$, reward function $R = (I - \gamma P)V$, and basis $\Phi$

where

$$\Phi = \begin{bmatrix} 1 & 0 \\ 0 & -1. \\ {}^1\!/_2(1.05 + \epsilon) & -{}^1\!/_2(1.05 + \epsilon) \end{bmatrix} \tag{25}$$

The basis includes the representation error term $\epsilon = 10^{-4}$.

For illustration purposes, we select the family of distributions $\mu = ({}^p\!/_2, {}^p\!/_2, 1 - h)$ parameterized by $p \in [0, 1]$. This characterizes the possible distributions of data that we will present to POP-QL and naive TD in this experiment. The on-policy distribution corresponds to $p = 0.5$. The contraction mapping condition is satisfied for the left subset of sampling distributions where $p \lesssim 0.51$ and not satisfied for the right subset where $p \lesssim 0.55$. This is immediately apparent in Fig. 4, where we plot the error at convergence from running naive- and POP-QL above, and the effective distribution of TD updates after reweighing. In the left subset, where the NEC holds, POP-QL does not reweight TD updates at all. Therefore, the error of POP-TD tracks that of naive TD, and the effective distribution of TD updates in POP-TD and naive TD are the same as the data distribution.

Fig. 4 also plots TD-Learning traces for three different sampling distributions using both Iterivate TD and POP-QL. We can see when $p = 0.8$ (right), the contraction mapping condition is violated Itertive TD diverges. However, POP-QL reweights the sampling distribution, yielding a new sampling distribution that satisfies the contraction mapping condition. Thus, POP-QL still converges in this example.

# B    Proofs and Derivations

## B.1    Derivation of Contraction Mapping LMI

For a finite MRP, the projected Bellman equation can be written in matrix notation as:

$$\mathcal{B}(\Phi w) = \Pi_\mu (R + \gamma P \Phi w), \tag{26}$$

By the definition of a contraction mapping, the projected Bellman equation is a $\gamma$-contraction mapping if and only if for all $w_1, w_2 \in \mathbb{R}^k$:

$$\|\mathcal{B}(\Phi w_1) - \mathcal{B}(\Phi w_2)\|_\mu \leq \gamma \|\Phi w_1 - \Phi w_2\|_\mu. \tag{27}$$

Now, consider the left side of this equation. We can rewrite this as follows:

$$\begin{aligned}
\|\mathcal{B}(\Phi w_1) - \mathcal{B}(\Phi w_2)\|_\mu &= \|\Pi_\mu \mathcal{B}(\Phi w_1) - \Pi_\mu \mathcal{B}(\Phi w_2)\|_\mu \\
&= \|\Pi_\mu(R + \gamma P \Phi w_1) - \Pi_\mu(R + \gamma P \Phi w_2)\|_\mu \\
&= \gamma \|\Pi_\mu P \Phi w_1 - \Pi_\mu P \Phi w_2\|_\mu \\
&= \gamma \|\Pi_\mu P \Phi \tilde{w}\|_\mu
\end{aligned}$$

where $\tilde{w} = w_1 - w_2$. Thus, the projected Bellman equation is a contraction mapping if and only if for all $w \in \mathbb{R}^k$,

$$\|\Pi_\mu P \Phi w\|_\mu \leq \|\Phi w\|_\mu. \tag{28}$$

Plugging in the closed form solution to the projection, we get,

$$\begin{aligned}
& w^T \Phi^T P^T D_\mu \Phi \left(\Phi^T D_\mu \Phi\right)^{-1} \Phi^T D_\mu \Phi \left(\Phi^T D_\mu \Phi\right)^{-1} \Phi D_\mu P \Phi^T w \leq w^T \Phi^T D_\mu \Phi w \\
\Leftrightarrow \quad & w^T \left(\Phi^T P^T D_\mu \Phi \left(\Phi^T D_\mu \Phi\right)^{-1} \Phi D_\mu P \Phi^T - \Phi^T D_\mu \Phi\right) w \leq 0 \\
\Leftrightarrow \quad & \Phi^T P^T D_\mu \Phi \left(\Phi^T D_\mu \Phi\right)^{-1} \Phi D_\mu P \Phi^T - \Phi^T D_\mu \Phi \preceq 0
\end{aligned}$$

Finally, using Schur Complements, we can convert this to an LMI,

$$F_\mu \equiv \begin{bmatrix} \Phi^T D_\mu \Phi & \Phi^T D_\mu P \Phi \\ \Phi^T P^T D_\mu \Phi & \Phi^T D_\mu \Phi \end{bmatrix} \succeq 0. \tag{29}$$

Thus, as long as $F_\mu \succeq 0$, the projected Bellman equation is a contraction mapping.

## B.2 Proof of Lemma 1

*Proof.* To prove this, we will use a simple reduction to a Markov chain.

For a fixed policy, $\pi$, our MDP, $\mathcal{M} = (\mathcal{S}, \mathcal{A}, P, R, \gamma)$ reduces to a MRP. We can write this new MRP as $\mathcal{M}^\pi = (\mathcal{X}, P^\pi, R, \gamma)$ where $\mathcal{X} = \mathcal{S} \times \mathcal{A}$ is the Cartesian product of the state and action spaces of the MDP and $P^\pi : \mathcal{X} \times \mathcal{X} \mapsto \mathcal{R}_+$ is defined as follows:

$$P^\pi((s,a),(s',a')) := p(s'|s,a)\pi(a'|s') \qquad \forall \, (s,a),(s',a') \in \mathcal{X}$$

We can see clearly that for all $(s,a) \in \mathcal{X}$, $\sum_{(s',a') \in \mathcal{X}} P^\pi((s,a),(s',a')) = 1$.

Since our MDP, $\mathcal{M}$, is finite, we can define the feature matrix, $\Phi \in \mathcal{R}^{n,k}$, using the MDP feature function, $\Phi_i = \phi(s_i, \pi(s_i))$ for each $s_i \in \mathcal{S}$.

Now, applying Theorem 2 from Kolter (2011) to our MRP, $\mathcal{M}_\pi$, we have that:

$$\|\Phi w^\star - V^\pi\|_\mu \leq \frac{1 + \gamma\sqrt{\delta(\nu,\mu)}}{1-\gamma}\|\Pi_\mu V^\pi - V^\pi\|_\mu \tag{30}$$

where $w^\star$ is the unique fixed point of Eq. (2), $V^\pi$ is the unique fixed point of $V^\pi = R + \gamma P^\pi V^\pi$, $\nu$ is the stationary distribution, and $\delta(\nu,\mu) = \max_{x,\tilde{x}} \frac{\nu(x)}{\mu(x)} \cdot \frac{\mu(\tilde{x})}{\nu(\tilde{x})}$.

Mapping this bound back onto the MDP, $\mathcal{M}$, yields the stated bound. $\qquad \square$

## B.3 Derivation of POP-QL Updates:

Here we will detail the derivation of the POP-QL gradient updates. The methods here follow those of the MRP derivation. To start, we can rewrite Eq. (23) as:

$$\underset{\pi,q}{\text{maximize}} \; \frac{1}{\beta}\mathbb{E}_\mu[Q^\pi(s,a)] - D_{\mathrm{KL}}(q \,\|\, \mu) \quad \text{s.t.} \quad \mathbb{E}_{(s,a)\sim q}[F^\pi(s,a)] \succeq 0 \tag{31}$$

Next, we introduce Lagrange variables to convert this into an unconstrained optimization problem:

$$\underset{q,\pi}{\text{maximize}} \, \underset{Z \succeq 0}{\text{minimize}} \; \frac{1}{\beta}\mathbb{E}_\mu[Q^\pi(s,a)] - D_{\mathrm{KL}}(q\|\mu) + \mathrm{tr}\, Z^\top \mathbb{E}_q[F^\pi(s,a)]$$

$$= \underset{\pi}{\text{maximize}} \left( \frac{1}{\beta}\mathbb{E}_\mu[Q^\pi(s,a)] + \underset{q}{\text{maximize}} \, \underset{Z \succeq 0}{\text{minimize}} \; -D_{\mathrm{KL}}(q\|\mu) + \mathrm{tr}\, Z^\top \mathbb{E}_q[F^\pi(s,a)] \right)$$

Now, we focus on the inner optimization problem over $q$ and $Z$. As in the MRP version, we assume that strong duality holds in this inner optimization problem. Under this assumption, the inner optimization problem can be equivalently written as:

$$\underset{Z \succeq 0}{\text{minimize}} \, \underset{q}{\text{maximize}} \; -D_{\mathrm{KL}}(q\|\mu) + \mathrm{tr}\, Z^\top \mathbb{E}_q[F^\pi(s,a)] \tag{32}$$

This problem can be solved as before. First, we solve for the analytical solution of $q$,

$$q^\star(s,a) \propto \mu(s,a)\exp\left(\mathrm{tr}\, Z^{\pi\top} F(s,a)\right). \tag{33}$$

Next, we plug this solution back into our inner optimization problem:

$$\underset{Z \succeq 0}{\text{minimize}} \; \log\left(\mathbb{E}_\mu\left[\exp\left(\mathrm{tr}\, Z^\top F^\pi(s,a)\right)\right]\right) \tag{34}$$

Now, using the reparameterization of $Z$ in Eq. (11), the full optimization problem (Eq. (31) becomes:

$$\underset{\pi}{\text{maximize}} \left( \frac{1}{\beta}\mathbb{E}_\mu[Q^\pi(s,a)] + \underset{A,B}{\text{minimize}} \; \log\left(\mathbb{E}_\mu\left[\exp\left(\|y_A\|_2^2 + \|y_B\|_2^2 + 2\mathbb{E}_{s'\sim p(\cdot|s,a),a'\sim\pi(\cdot|s')}\langle y_B, y_A'\rangle\right)\right]\right)\right) \tag{35}$$

where $y_A = A^\top \phi_{\theta\phi}(s,a)$, $y_B = B^\top \phi_{\theta\phi}(s,a)$, and $y_A' = A^\top \phi_{\theta\phi}(s',a')$.

Again, we need to perform a 2-timescale gradient descent for $A, B$ since we are approximating an expectation inside of a exponential. Thus, we also learn a parameterized function $g_\theta$ to approximate the following:

$$g_\theta(s, a) \approx \mathbb{E}_{s' \sim p(s'|s,a), a' \sim \pi(s')} \left[ \langle y_B, y'_A \rangle \right] \tag{36}$$

Using this approximation, the gradient of $A$ and $B$ can be expressed as:

$$\mathbb{E}_{s,a \sim \mu, s' \sim p(s'|s,a), a' \sim \pi(\cdot|s')} \left[ \begin{array}{c} \exp\left(\|y_A\|_2^2 + \|y_B\|_2^2 + 2g_\theta(s,a)\right) \\ \cdot \nabla_{A,B} \left(\|y_A\|_2^2 + \|y_B\|_2^2 + 2\langle y_B, y'_A \rangle\right) \end{array} \right] \tag{37}$$

Next, we derive the policy updates using both Lagrange variables.

$$\nabla_\pi \left( \mathbb{E}_\mu[Q^\pi(s,a)] + \beta \log \left( \mathbb{E}_{\mu,\pi} \left[ \exp\left(\|y_A\|_2^2 + \|y_B\|_2^2 + 2\langle y_B, y'_A \rangle\right) \right] \right) \right)$$
$$\approx \nabla_\pi \mathbb{E}_\mu[Q^\pi(s,a)] + \beta \exp\left(\|y_A\|_2^2 + \|y_B\|_2^2 + 2g_\theta(s,a)\right) \nabla_\pi \mathbb{E}_\pi \left[ \|y_A\|_2^2 + \|y_B\|_2^2 + 2\langle y_B, y'_A \rangle \right]$$
$$= \nabla_\pi \mathbb{E}_\mu[Q^\pi(s,a)] + 2\beta \exp\left(\|y_A\|_2^2 + \|y_B\|_2^2 + 2g_\theta(s,a)\right) \nabla_\pi \mathbb{E}_\pi \left[ \langle y_B, y'_A \rangle \right]$$

Finally, the Q-learning reweighting terms are simply:

$$u(s, a) \approx \exp\left(\|y_A\|_2^2 + \|y_B\|_2^2 + 2g_\theta(s,a)\right) \tag{38}$$

# C    POP-QL Details

Here we provide additional details for the POP-QL algorithm.

## C.1    I- and M-Projections

The information (I-) projection and moment (M-) projection are defined as follows:

$$\text{I-Projection: } \min_q D_{\text{KL}}(q\|\mu) \qquad \text{M-Projection: } \min_q D_{\text{KL}}(\mu\|q) \tag{39}$$

Since the KL-divergence is an asymmetric measure, these projections are usually not equivalent. A key difference between the I- and M-projections is that the I-projection tends to under-estimate the support of the fixed distribution $\mu$, resulting in more density around the modes of $\mu$, while the M-projection tends to over-estimate the support of $\mu$, resulting in a higher variance solution.

In the context of off-policy Q-learning, sampling states and actions with very low or zero support under the sampling distribution, $\mu$, can result in over-estimating the Q-function, which in-turn results in a poor performing policy. For this reason, we argue the more conservative I-projection is a better fit for off-policy Q-learning.

## C.2    g-function Normalization

In practice, we found normalizing the $g$ function by the spectral norm of the $A$ and $B$ matrices improved learning stability. Specifically, we train the $g$ network to approximate the following quantity:

$$g_\theta(s, a) \approx \frac{1}{\|A\|_2 \|B\|_2} \mathbb{E}_{s' \sim p(s'|s,a), a' \sim \pi(s')} \left[ \langle B^\top \phi(s,a), A^\top \phi(s',a') \rangle \right] \tag{40}$$

where $\|A\|_2$ represents the spectral norm of $A$. Note that, by definition of the spectral norm and since $\|\phi(s,a)\|_2 = 1$ for all $s$, this bounds the range of the $g$ function, $g_\theta(s, a) \in [-1, 1]$.

## C.3    Hyper-Parameter Search

### Q-Function and Policy Learning Rates

We looked at using $\lambda_Q, \lambda_\pi =$, 3e-4, 1e-4, 3e-5, and 1e-5. The lower learning rates for the policy seemed to significantly improve asymptotic performance of our method. However, when setting

$\lambda_\pi = $ 1e-5, we found it took too long to learn a decent policy. Thus, for our experiments, we chose the middle-ground of $\lambda_\pi = $ 3e-5. We found $\lambda_Q = $ 1e-4 worked the best for POP-QL.

**Lagrange Matrices Learning Rate**

Since we want to learn the Lagrange matrices assuming a fixed policy, we used an increased learning rate for the Lagrange matrices compared to the policy. We tried

**g-Function Learning Rate**

Once we normalized the g-function (as described in Appendix C.2), we found the performance of the algorithm seemed to be quite robust to the choice of g-function learning rate. We tried 1, 10, and 20 times the learning rate of the Lagrange Matrices and all performed roughly equally. We used $\lambda_g = 10\lambda_{[A,B]}$ for our experiments.

**KL-Q-value weighting parameter $\beta$**

This $\beta$ term weights how much POP-QL's weights policy performance versus the KL-divergence between the new sampling distribution and the reweighted distribution. The larger $\beta$ is, the more the policy is projected and the less significant the reweighting terms become.

In the D4RL problems, we need a large $\beta$ term to make sure the policy does not drive the agent too far outside the data distribution. We tested $\beta = $ 100, 30, 10, 3, 1, and 0.3. Since our choice of $\beta$ depends on the estimated Q-values, we chose a different $\beta$ beta for each class of domains. For both the Mujoco and Franka Kitchen domains, we chose $\beta = 100$.

## C.4 Target Networks

Just as in other actor-critic methods, we use a target network for the features and Q-function weight $w$ to stabilize the Q-learning updates.

We also tested using these target features for training Lagrange matrices $A$ and $B$, but found this reduced performance. So instead, we do not use any target networks in training the Lagrange matrices.

# D Additional Experiment Details

## D.1 Frozen Lake

### D.1.1 Features

To construct the features for this domain, we constructed a random vector of dimension $k = 60$ for each state-action pair. This vector was sampled from a uniform distribution $\mathcal{U}([0, 1]^k)$, then normalized to have a unit norm.

### D.1.2 Hyper Parameter Tuning

We performed a cursory hyper-parameter search for this problem. Online SAC has little problem converging and, thus, is not sensitive to hyper-parameters. In the offline case, when the dimension of the features, $k$, is greater than or equal to the size of the state-action space (64), offline SAC is also very stable, since this reduces to tabular RL. As soon as $k < |S||A|$, however, offline SAC becomes very unstable and sensitive to hyper-parameters.

For all methods, we used a Q-function learning rate of $1 \times 10 - 3$ and a policy learning rate of $1 \times 10 - 4$. We found that this lead to convergent behavior for both CQL and POP-QL.

All methods also used SAC with automatic entropy tuning. Through a course search, we found a target entropy of 0.5 worked well across CQL and POP-QL.

Both CQL and POP-QL have regularization parameters that keep the policy close to the data-collection policy. These parameters improve convergence at the expense of performance. We tuned this parameters by picking the lowest values that still lead to convergent behavior. For CQL, this was $\alpha = 0.5$. For POP-QL, we useed $\beta = 0.5$.

## D.2   D4RL

### D.2.1   Hyper Parameter Tuning

As mentioned in the main paper, using the hyper-parameters presented in Kumar et al. (2020b), we were not able to reproduce the results of CQL presented in the paper. Specifically, setting Lagrange parameter $\tau$ to the value suggested in the paper resulted in very poor performance. Instead we did a cursory hyper-parameter sweep using $\tau = 3$, $1$, $0.3$, and $0.1$. For MuJoCo, we found $\tau = 0.3$ peformed best. For kitchen, we found $\tau = 3$ peformed best.

### D.2.2   Network Structure

For each baseline method, we use a 2 hidden fully connected neural network with a width of 256 and ReLU activations for both the policy network and Q-network. For the g-network for POP-QL, we use a 4 hidden layer network with a width of 1024 and ReLU activations.

