# OpenReview forum: "Projected Off-Policy Q-Learning (POP-QL) for Stabilizing Offline Reinforcement Learning"
_ICLR.cc/2024/Conference — Submitted to ICLR 2024_

### Official Review · Reviewer_JhFx · 2023-10-30

**Soundness:** 3 good
**Presentation:** 3 good
**Contribution:** 2 fair
**Rating:** 5
**Confidence:** 3

**Summary:**

The paper discusses the challenges and potential solutions associated with off-policy reinforcement learning (RL) using Temporal Difference (TD) learning. The authors propose a new approach called Projected Off-Policy Q-Learning (POP-QL). The key contributions of POP-QL are twofold:

Computational Efficiency: POP-QL introduces a new sampling projection that is computationally efficient because it provides a closed-form solution for a specific optimization problem.

Improved Q-Learning Algorithm: The algorithm jointly projects the policy and sampling distribution, which can be integrated into any Q-learning algorithm. This reduces the approximation error of the Q-function, enhancing policy performance.

The paper presents the technical details of the POP-QL method, focusing on its implementation in Markov Reward Processes (MRP) and its extension to Markov Decision Processes (MDP). The method centers on the contraction mapping condition and offers computational advantages over prior approaches, making it suitable for large-scale applications.

**Strengths:**

The authors delve deep into the mathematical foundations of their approach, from exploring the contraction mapping condition to presenting solutions for the dual problem. Such rigorous theoretical detail ensures that their approach is grounded in solid mathematical principles.

**Weaknesses:**

1. Literature Review Limitation:

The paper only reviews literature up to 2022, with a single paper after that year mentioned. Given the rapid evolution in the field of offline RL, this makes the literature review less comprehensive. New techniques, challenges, or advancements in offline RL post-2022 might have been overlooked, making the foundation of the paper potentially outdated.

2. Incremental Framework:

The framework is described as an incremental advancement on Kolter (2011). An incremental approach might not provide significant improvements or breakthroughs over existing methodologies.

3. Poor Evaluation Results:

While the paper presents Projected Off-Policy Q-Learning (POP-QL) as a novel method with certain improvements over the previous works, the evaluation results indicate that it struggles to match or surpass state-of-the-art performance in scenarios with near-expert data-collection policies. This raises questions about its practical utility and whether the theoretical improvements translate to real-world benefits.

**Questions:**

How generalizable is the contraction theory discussed in this paper to other offline algorithms that use a structure of Soft Actor-Critic (SAC) like Conservative Q-Learning (CQL)? Could the author provide an ablation study to test the effect of different components (including the specific design in Section 4.3 and the policy optimization) on the performance of the algorithm?

Will there be some other datasets that the algorithm can perform better than other SOTA methods? The authors mentioned that this algorithm performs better when the datasets are far from the optimal policy distribution, specifically the “random” datasets. However, this method still performs worse than other method in walker2d-random. Will the author generate some new datasets using some random policies in the mujoco environment and test the algorithm's performance?

---

> ### Author Response · Authors · 2023-11-22
>
> Thank you for your review and feedback.
>
> Regarding the “incremental advancement” point, we strongly disagree with this point. We do share the motivation and use of the contraction mapping condition with Kolter (2011), but our approach differs greatly after that. Firstly, the approach in Kolter (2011) only works for very small MRPs. Our approach uses a completely new technique for satisfying the contraction mapping condition and includes a policy projection step that is crucial for policy optimization. We will make sure to emphasize this further.
>
> Regarding the poor performance results, what is unique about our method is that, unlike the state-of-the-art offline RL techniques, our method does not regularize the policy towards the data policy. For that reason, we expect conservative methods and explicit policy regularization methods to out-perform our method on near-optimal datasets. In our (updated) small-scale results and the “random” D4RL results we show that as the dataset becomes less optimal, our method outperforms other methods. We will try to make this point clearer.
>
> Your questions regarding additional datasets is a great point. We tried to stick with common benchmark datasets because we didn’t want the datasets to seem contrived. But adding additional datasets might further illustrate that point.
>
> Regarding the ablation study question, this is a good suggestion. From our informal tests, without the policy projection, POP-QL diverges and when combining CQL with POP-QL we get some slight improvements, but we will conduct these experiments formally and add them to the results.
>
> Finally, regarding the lack of more recent literature, we will make sure to include some of the most recent publications in the space.

---

### Official Review · Reviewer_Urdi · 2023-10-31

**Soundness:** 2 fair
**Presentation:** 3 good
**Contribution:** 2 fair
**Rating:** 5
**Confidence:** 3

**Summary:**

This paper revisited the idea of Kolter's work [1], that consider a relaxed contraction constraint $||\Pi_D P \Phi w||_D \leq ||\Phi w||_D$, where ``$\Pi_D$ projects the transition matrix $P$ onto the bases will be non-expansive for any function already in the span of $\Phi$", to address the divergence problem of off-policy TD methods.  The authors further leverage strong duality and an analytical solution for the optimization variable $q$ (reweighted distribution) to reduce computational costs, enabling scalability to complex environments.

[1] Kolter, J. "The fixed points of off-policy TD." Advances in neural information processing systems 24 (2011).

**Strengths:**

- The paper is well-written, with a well-organized related work section and clearly articulated motivations.

- The application of off-policy TD methods to offline RL is an interesting and potentially fruitful direction.

- The algorithmic development appears sound.

**Weaknesses:**

- Empirical assessment is limited:

    - POP-QL seems to surpass (regularization-based) baseline methods only in hopper and halfcheetah random tasks. Additionally, as shown in Figure 3, CQL consistently demonstrates superior performance irrespective of the off-policy data ratio. This raises my reservation regarding the effectiveness of the proposed method.


    - It could be insightful to delve deeper into the reduction of value-approximation error. This would highlight the merits of using the contraction constraint in offline RL.  Additional demonstrations highlighting POP-QL's capability in reducing value-error would be highly valuable. ([2] could be an example of illustrating value approximation error in deep RL.)

    - (Minor point) An ablation study on $\beta$ would be beneficial.

[2] Fujimoto, Scott, et al. "Why should i trust you, bellman? the bellman error is a poor replacement for value error." arXiv preprint arXiv:2201.12417 (2022).

**Questions:**

While the paper presents its motivation and algorithm development effectively, and I appreciate its potential value to the offline RL community, the current version might benefit from some additional discussion, especially in its empirical evaluation. Demonstrating its advantage and effectiveness may require a more thorough assessment.

---

> ### Author Response · Authors · 2023-11-22
>
> Thank you for your review.
>
> Regarding the Frozen Lake performance, hopefully our updated results illustrate the point better.
>
> Regarding your question about value-approximation error, the challenge here is that except in small tabular cases, we cannot accurately estimate the true value function. Thus, any approximation of true “value-error” is only as good as our function approximator. Do you have any suggestions on how to illustrate a reduction in approximation error?

---

### Official Review · Reviewer_Mmg3 · 2023-11-05

**Soundness:** 2 fair
**Presentation:** 2 fair
**Contribution:** 2 fair
**Rating:** 3
**Confidence:** 4

**Summary:**

This paper proposes an offline RL algorithm called POP-QL based on Kolter (2011). POP-QL reweighs the data distribution so that off-policy TD can be run stably offline. Compared with the original idea in Kolter (2011) based on LMI, the authors show that POP-QL can more be more easily scaled. The authors provide a practical implementation of POP-QL and test it empirically in a toy off policy evaluation problem (frozen lake) and the d4rl benchmarks.

**Strengths:**

This paper takes a different approach to offline RL from other existing works. To my knowledge, this approach is novel and is interesting to learn.

**Weaknesses:**

The derivation of the method on the theoretical side is incomplete. In addition, there's no formal guarantee on the performance of the learned policy. The empirical evaluation is also quite limited. The proposed algorithm has pretty weak empirical performance, and the baselines compared here do not fully reproduce what's been reported in the literature. Therefore, at this point, it is unclear whether the proposed idea actually works. Overall, while the idea is interesting, I think more works are needed in both improving the theoretical and empirical sides.

**Questions:**

1. In (7), should q be subject to a distribution constraint. i.e. q>=0 and \sum q = 1?

2. The derivation of Lemma 1 is incomplete. It uses Theorem 1, which is not given.

3. A derivation bug? In (9), it shows q* is proportional to things on the right, not equal. The missing normalization factor there while not a function of (s,a) is a function Z. Therefore, (10) is not exactly equal to (8). (Same issue for (18) later on).

4. In (24), how is \nabla_\pi E_\mu[ Q^\pi(s,a)] computed? how do you get Q^\pi (which cannot be estimated in general for offline RL) and how is its gradient to \pi computed?

5. The empirical results are pretty weak, so I'm not sure if it shows enough evidence that the proposed method is working. For mujoco tasks, except random datasets, the rests are way below what's established in the literature.

6. The baselines' performance is not reproducing what's known in the literature (I'm not asking for precisely getting the same numbers but now it's pretty far from the typical ballpark.) E.g. CQL has been reproduced by many papers with roughly 10%-20% differences from the original paper's results (see e.g. (Li et al. 2023) as a recent example). Not being able to reproduce existing results make the experimental results questionable.

Ref: Li, et al. "Survival Instinct in Offline Reinforcement Learning." arXiv preprint arXiv:2306.03286 (2023).


Minor:
In (7) D_KL = KL?

---

> ### Author Response · Authors · 2023-11-22
>
> Since we state that $q$ is a distribution, we chose not to explicitly state that $q>=0$ and $\sum q = 1$ since we thought that was implied. But we can add that in if it makes it clearer.
>
> Good catch on the proof of Lemma 1! This is meant to refer to Theorem 2 from Kolter (2011). In a previous draft of the paper, we restated this theorem, which is why there was a missing reference.
>
> Regarding the derivation bug, we do in fact keep track of the normalization term, even though we don’t explicitly state it in the paper. In fact, ignoring the normalization term leads to a very different result. Below is our step-by-step derivation. We’ll try to make this clearer.
>
> Your point about the baselines is a good point and something we struggled with. The reason is around the “Lagrange threshold” parameter discussed in the Appendix F of Kumar et al. (2020). The goal of this parameter is to automatically tune the conservatism parameter, \alpha, to the specific task (similar to the automatic entropy tuning of SAC). However, when using the library provided by the authors (https://github.com/young-geng/JaxCQL), using the automatic \alpha tuning significantly reduces performance for some of the tasks compared to a fixed \alpha. Reading through the paper you sent, it looks like Li et al. (2023) used a fixed \alpha parameter, which is different from the parameters reported in Kumar et al. (2020). For our next revision, we will re-run the CQL experiments using a fixed \alpha parameter and report both numbers.
>
> Regarding your question on equation 24, this is the gradient with respect to the Q-network not the true Q-function. We tried to make the equation less cluttered by removing the parameter subscript, but that clearly made it more confusing. We will make that clearer.
>
> Thanks for the note on using two different notations for KL. We fixed that in the newest version.

---

> ### Author Response · Authors · 2023-11-22
> **Derivation of equation 9**
>
> Our analytical solution (equation 9) can be written as:
>
> $q^{\star}(s) = \frac{1}{C} \ \mu(s) \exp \left(\operatorname{tr} Z^T F(s)\right)$
>
> With $C = \sum_s \mu(s) \exp \left(\operatorname{tr} Z^T F(s)\right)$.
> Plugging this into equation 8 yields.
>
> $\max_{Z \succeq 0} \mathrm{KL}(q^* \| \mu)-\operatorname{tr} Z^T \mathbf{E}_{q^*}[F(s)]$
>
> $\max_{Z \succeq 0} \ \mathbf{E}_{q^*} [ \log \left(\frac{1}{C} \frac{q^*(s)}{\mu(s)} \right) ] - \operatorname{tr} Z^T \mathbf{E}{q^*} \left[ F(s) \right]$
>
> $\max_{Z \succeq 0} \ \mathbf{E}_{q^*} \left[- \log(C) + \log \left( \exp \left(\operatorname{tr} Z^T F(s)\right) \right) \right] -\operatorname{tr} Z^T \mathbf{E}{q^*}[F(s)]$
>
> $\max_{Z \succeq 0} \ - \log(C)$
>
> $\max_{Z \succeq 0} \ - \log\left(\mathbf{E}_{\mu} \exp \left(\operatorname{tr} Z^T F(s)\right)\right)$

---

### Official Review · Reviewer_5Wu7 · 2023-11-08

**Soundness:** 2 fair
**Presentation:** 3 good
**Contribution:** 2 fair
**Rating:** 3
**Confidence:** 4

**Summary:**

This paper presents a method for tackling the distribution shift that arises between the policy distribution used for data collection and the current policy (distribution), building upon the work of Kolter (2011). The authors introduce a computationally more efficient approach based on Kolter's method that is applicable to deep reinforcement learning (RL) settings to some extent by projecting the policy and sampling distributions. The effectiveness of their method is evaluated through limited experiments conducted primarily in offline RL settings.

J Kolter. The fixed points of off-policy td. Advances in Neural Information Processing Systems, 24: 2169–2177, 2011.

**Strengths:**

The paper is written well (except in a few cases) and studies an important problem in RL. The proposed method is more computationally efficient than the original Kolter's method and has better applicability in (deep) RL.

**Weaknesses:**

- Although this paper tackles an important problem in RL, the experimental results fail to demonstrate any advantages gained from employing this method in offline RL. Moreover, this method introduces more complexities compared to other offline RL methods and has more moving parts which further diminishes its appeal in light of the performance of other offline methods which outperform this method.

- While the distribution shift problem in offline RL is a very important issue like online RL, offline RL also deals with extrapolation problem and limited data availability which makes it an even harder problem than online-RL. As evidenced by the results of this paper, methods like the one proposed in this paper struggles to perform effectively in offline setting. It is likely that such methods would be better suited for online RL, particularly in off-policy setting. But the paper doesn't consider off-policy setting.

- The paper fails to adequately discuss relevant off-policy methods that deal with distribution mismatch such as [1,2] among many others. In particular, this paper should have discussed and compared with existing off-policy RL methods.

- The scope of experiments are very limited and it should contain more benchmarks ( from online RL) and baseline methods,

- (minor issue ) The paper conflates offline RL with off-policy RL in many places. For example, in page 3, "In the off-policy setting, we assume the agent cannot directly interact with the environment". There are examples like this in the paper. Note that in off-policy, the agent still interacts with the environment, but the policy that is being used to collect data differs from the policy being optimized. In contrast, offline-RL is the one in which no further interaction (beyond data collection phase) with the environment is allowed.

- Authors mentioned in page 3 that "One critical challenge with IS methods is that they do not address support mismatch and, thus, tend to perform poorly on larger scale problems." That is not entirely correct as the main objective of importance sampling is to alleviate distribution mismatch by learning unbiased estimators, albeit at the expense of increased variance.

Given the restricted results, experiments, and applicability of the proposed method, the paper requires major revision before being considered for publication at ICLR and it is not ready for ICLR. I'd recommend authors to consider the comments and resubmit the paper once it is ready.

[1] P3O: Policy-on Policy-off Policy Optimization (https://proceedings.mlr.press/v115/fakoor20a.html)

[2] Experience Replay with Likelihood-free Importance Weights (https://proceedings.mlr.press/v168/sinha22a.html)

**Questions:**

N/A

---

> ### Author Response · Authors · 2023-11-22
>
> Thank you for your review.
>
> Thank you for pointing out the missing literature. We agree we should spend more time discussing other off-policy methods and will make sure to include those papers.
>
> And thank you for pointing out mistakes regarding mixing up off-policy and offline RL. These are typos and we will fix them.
>
> The reason we chose to evaluate the fully offline setting is because the challenges of distribution shift are more pronounced. Unlike our method, all the best performing offline RL methods have some regularization back to the data policy. For that reason, of course our method will not be able to compete with policy-regularization or conservative methods on near-optimal datasets. However, our (updated) small scale results and the “random” D4RL results illustrate that this method can outperform policy-regularization and conservative methods when the datasets are severely sub-optimal. We will try to further highlight this.
>
> Regarding your point about IS methods, our understanding is most IS methods (like many other off-policy methods) assume the support of the data and candidate policies match, otherwise there would be division-by-zero issues. However, in the offline RL case, this assumption is often violated. Could you elaborate on what part you disagree with?

---

### Author Response · Authors · 2023-11-22

We would first like to thank the reviewers for the time and effort you all put into these reviews. Given the low scores, it seems unlikely that this version of the paper will get accepted. However, we will still hope to answer all your questions and respond to the criticisms and would really appreciate your feedback.

We’ve updated the paper with new results for the small-scale Frozen Lake experiment, which were unfortunately not finished by the time of submission. The reason for the difference was a small bug in the environment setup.

---

### Meta-Review · Area_Chair_52AH · 2023-12-07

**Metareview:**

The paper proposes a reweighing mechanism for off-policy RL, where pessimism or behavioral regularization are typically used.

However, the method is more complex than prior art without clearly outperforming it, and it is not backed by substantial theory.
The experiment are also somewhat limited. Unfortunately, the reviewers recommend rejection.

**Justification For Why Not Higher Score:**

More complex algorithm with unclear benefits

**Justification For Why Not Lower Score:**

N/A

---

### Decision · Program_Chairs · 2024-01-16

Reject